# Magnetic resonance imaging analysis predicts nanoparticle concentration delivered to the brain parenchyma

Michael Plaksin [1,3✉], Tiran Bercovici[1,3], Gabriella Gabi Sat Toltsis[1], Javier Grinfeld[1], Boaz Shapira[1], Yuval Zur[1], Rafi de Picciotto[1], Eyal Zadicario[1], Mustaffa Siddeeq[2], Anton Wohl[2], Zion Zibly[2], Yoav Levy[1] & Zvi R. Cohen [2✉]

Ultrasound in combination with the introduction of microbubbles into the vasculature effectively opens the blood brain barrier (BBB) to allow the passage of therapeutic agents. Increased permeability of the BBB is typically demonstrated with small-molecule agents (e.g., 1-nm gadolinium salts). Permeability to small-molecule agents, however, cannot reliably predict the transfer of remarkably larger molecules (e.g., monoclonal antibodies) required by numerous therapies. To overcome this issue, we developed a magnetic resonance imaging analysis based on the $\Delta R_2^*$ physical parameter that can be measured intraoperatively for efficient real-time treatment management. We demonstrate successful correlations between $\Delta R_2^*$ values and parenchymal concentrations of 3 differently sized (18 nm–44 nm) populations of liposomes in a rat model. Reaching an appropriate $\Delta R_2^*$ value during treatment can reflect the effective delivery of large therapeutic agents. This prediction power enables the achievement of desirable parenchymal drug concentrations, which is paramount to obtaining effective therapeutic outcomes.

[1] Insightec Company Ltd, Tirat Carmel, Israel. [2] Neuro Oncology Unit, Sheba Medical Center, Ramat Gan, Affiliated with the Sackler School of Medicine Tel Aviv University, Tel Aviv, Israel. [3] These authors contributed equally: Michael Plaksin, Tiran Berkovitz. ✉email: MichaelP@insightec.com; Zvi.Cohen@sheba.health.gov.il

Ultrasound induces a variety of non-thermal bioeffects, ranging from aggressive effects such as mechanical tissue ablation–histotripsy[1], to more delicate effects such as ultrasonic neuromodulation[2–4]. The safety and efficacy of ultrasound to open the blood brain barrier (BBB) for transmission of various neurologic therapeutics has been explored over the last two decades[5–7]. The BBB impedes the transmission of molecules with a molecular weight greater than 400 Da[8] from the blood to the brain parenchyma, thereby diminishing the effectiveness of a vast majority of neurotherapeutic agents and biomarkers. Ultrasound overcomes this limitation, demonstrating effective BBB-opening mediated by intravascular microbubble oscillations[9,10].

Overcoming the BBB for drug delivery is achieved by other technologies as well, including direct surgical injection[11–13], intranasal delivery[14–16], active efflux transporter-targeted strategies[17–19], tight junction-targeted strategies[20,21], and magnetic resonance-guided laser ablation techniques[22,23]. Ultrasound has emerged as a particularly promising technology, however, as it is noninvasive, has a millimetric scale level of precision, and leads to robust, consistent, safe, and reversible BBB-opening[24,25].

Ultrasound enables the transfer of different-sized molecules through the BBB, from 1-nm-sized molecules such as Omiscan, Gadavist, and Dotarem[26–28], to tens of nanometers-sized molecules, such as fluorescein-tagged dextrans[29,30], gold nanoparticles[31], adeno-associated viral vectors[32], and liposomal doxorubicin[5], and even neural stem cells several microns in size[33].

Despite achieving cross-BBB transfer of molecules/nanoparticles with a very wide range of sizes, previous studies generally present specific acoustic parameters that must be met for the effective transfer of different-sized molecules through the BBB. To the best of our knowledge, a robust solution to forecast the precise quantity of the therapeutic agent that will effectively reach the targeted area during and after the BBB-opening treatment has not been presented.

In the present study, we propose utilizing $\Delta R_2*$, a physical parameter that is regularly measured in magnetic resonance imaging (MRI), to measure and predict the effectiveness of BBB-opening during treatment. Although $\Delta R_2*$ is typically used as a safety measure to detect treatment-related petechiae during BBB-opening procedures[27,34], the utility of $\Delta R_2*$ is not limited to measuring vascular damage. The results of the present study demonstrated that $\Delta R_2*$ values clearly reflected the parenchymal concentrations of 3 clinically relevant sizes of liposomes that were conjugated to gadolinium (Gd) ions to make them visible and quantifiable in the MRI environment. Further, working in $\Delta R_2*$ mode is not necessarily accompanied by vascular damage[34].

Our results demonstrate the BBB transfer dynamics of different sized molecules and emphasize the problem of using small Gd-salts as a surrogate marker for BBB transfer of much larger molecules and drugs such as monoclonal antibodies and liposomal therapeutics, thereby highlighting the limitations of applying small Gd-salts for predicting effective therapeutic outcomes.

The present findings may pave the way toward enhanced and effective systemic delivery of drug-loaded lipid nanoparticles for controlled brain therapy following ultrasound-mediated BBB-opening. Finally, we believe that the proposed method creates the ideal setting for potential clinical trials and may become a valuable therapeutic modality for a variety of neurologic disorders, such as brain tumors, neurodegenerative diseases, and cognitive disorders.

## Results

All experiments were performed on male Sprague Dawley rats. The rats were placed on an MRI Guided Focused Ultrasound (MRgFUS) transducer (Exablate Model 4000 Type-2 system, Fig. 1a), which was then placed in the MRI, and treatment planning images were acquired to define the treatment area (Fig. 1b, see "Methods" for details).

$T_1$w, $R_2*$ and multi-echo $R_2*$ magnetic resonance (MR) images were acquired before and after the BBB-opening treatment and after injecting Gd-liposomes to determine the correlation between $\Delta R_2*$ values and the Gd-liposome concentration. Details of the experimental flow, MRI analysis flow, and MRI analysis basics are provided in the "Methods", and in Figs. 2 and 3, respectively.

**Liposome endurance in the parenchyma.** Before performing detailed analyses, we injected different liposome populations into different rats to detect their presence with MR $T_1$ images; during this exploration, we observed that liposome presence was related to changes in the local qualitative $R_2*$ values (Fig. 4a). This is reasonable as changes in $R_2*$ are associated with the extravasation of a large protein—deoxyhemoglobin[35,36]; the extravasation level of these molecules can be correlated with non-destructive BBB transfer of other large molecules[34,37] (see also "Discussion"). We then explored how long after injection the liposomes can be detected by MR and how long the liposomes remain in the brain parenchyma. This examination revealed that unlike low-molecular weight MR contrast agents like Dotarem and Gadavist (Gd-salts) that are traceable for several minutes after injection but are totally washed out from the parenchyma several hours later[38], the liposomes were detected in the brain beginning several hours after the injection and remained in the tissue for at least 1 month (Fig. 4b). This phenomenon can be explained by the size of the liposomes, which are an order of magnitude larger than the Gd-salts[39,40], thus complicating their transfer through the BBB, and by the tens of hours half-life of PEG-based liposomes in the blood[41], which allows the liposomes to accumulate in the parenchyma and slows down the washout process to final

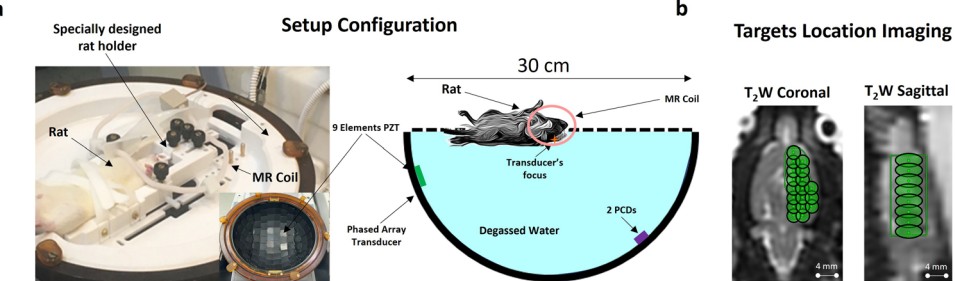

**Fig. 1 Experimental setup. a** MRgFUS semi-hemisphere transducer (diameter: 30 cm, 1024 elements) with a specially designed holder on top of the transducer and rat inside. The magnetic resonance receiving coil was placed on top of rat's head area. Two passive cavitation detectors captured the emitted pressure waves from microbubble activities. **b** $T_2$-weighted planning images with target locations. The targets covered the right hemisphere (not including the cerebellum) and contained 15–18 sub-spots with a distance of 2 mm between sub-spot centers.

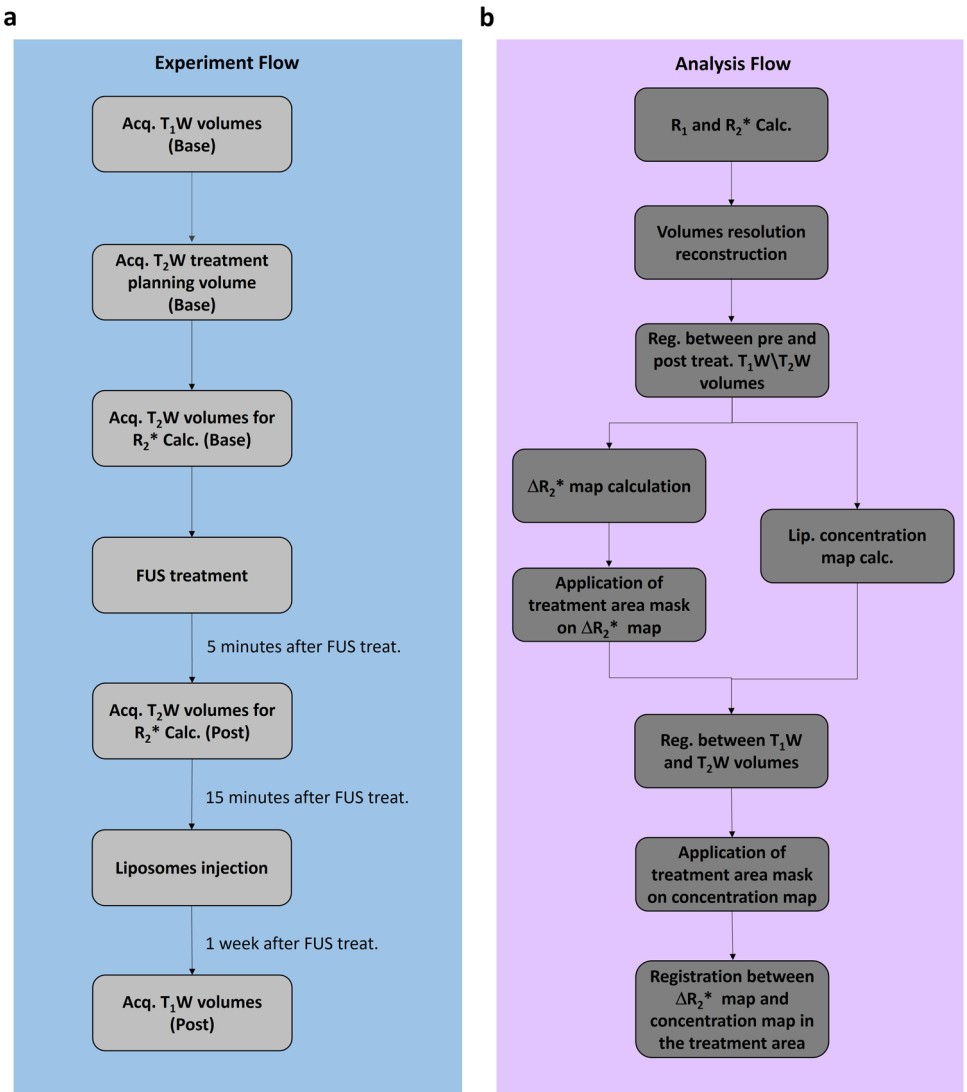

**Fig. 2 Experiment and MRI analysis flow. a** Experiment description: the experiments started with baseline $T_1$-weighted volume acquisition, followed by $T_2$-weighted images to place the treatment targets. Before FUS treatment, $T_2$-weighted GRE volumes were acquired to calculate baseline $R_2^*$ values. After the FUS treatment, GRE volumes were acquired again to calculate the post-treatment $R_2^*$ and, consequently, treatment-related $\Delta R_2^*$. The liposomes were injected ~15 min after the treatment, followed by $T_1$-weighted volume acquisition 1 week after the treatment. **b** MR images analysis flow: analysis started with pre- and post-treatment $R_2^*$ and $R_1$ volume calculations, followed by volume reconstruction to symmetric voxel $0.5^3$ mm$^3$. Then registration between pre- and post-treatment $T_2$-weighted volumes and pre- and post-treatment $T_1$-weighted volumes led to the $\Delta R_2^*$ and $\Delta R_1$ calculations. The $\Delta R_1$ value was converted to the Gd-liposome concentration using the calibration curves shown in Supplementary Fig. 2. The treatment mask was extracted from the $\Delta R_2^*$ map using the sub-spot location. After registration between $T_1$- and $T_2$-weighted volumes, treatment masks were applied to the Gd-liposome concentration volume to isolate concentration changes related to the treated area. To mimic the liposome diffusion, registration between the $\Delta R_2^*$ and Gd-liposomal concentration maps in the treated area was performed to find the geometric correlation between the two parameters.

stabilization of the parenchymal concentration following closure of the BBB.

**Gd-liposomal concentration can be predicted by $\Delta R_2^*$.** We then quantified the $\Delta R_2^*$ and Gd-liposomal concentrations in 18 rats to determine the correlation between them, as described in the "Methods". We first extracted the voxel-wise relation between the Gd-liposomal concentration and $\Delta R_2^*$ values at a single rat level. This analysis revealed higher Gd-liposomal concentrations with higher $\Delta R_2^*$ values, but the trend of same-sized liposomes varied between rats (Fig. 5a).

Next, we placed the data from all rats injected with the same liposome population in a single plot (Fig. 5b). The analysis revealed a correlation between the Gd-liposomal concentration

and $\Delta R_2^*$ value, with the parenchymal concentration of smaller liposomes showing a stronger dependence on the $\Delta R_2^*$. In addition, a size-dependent threshold was observed between the populations; when below the threshold ($\Delta R_2^* = 1.16$, 3.6, and 6.22 Hz, for 18-, 24-, and 44-nm liposomes, respectively), the effective liposome concentration had weak to no dependence on $\Delta R_2^*$, and above the threshold, the analysis showed monotonous increasing dependence. One-way analysis of variance (ANOVA, $a = 0.05$) between the parenchymal concentration of the three liposome populations in 2-Hz $\Delta R_2^*$ bin intervals revealed a statistically significant difference between the three populations in the $\Delta R_2^*$ range of 4–10 Hz (highest $p$ value $= 1.28 \times 10^{-8}$); the same statistically significant difference was also detected in the 2-Hz $\Delta R_2^*$ bin intervals between 18-nm liposomes and 24-nm or

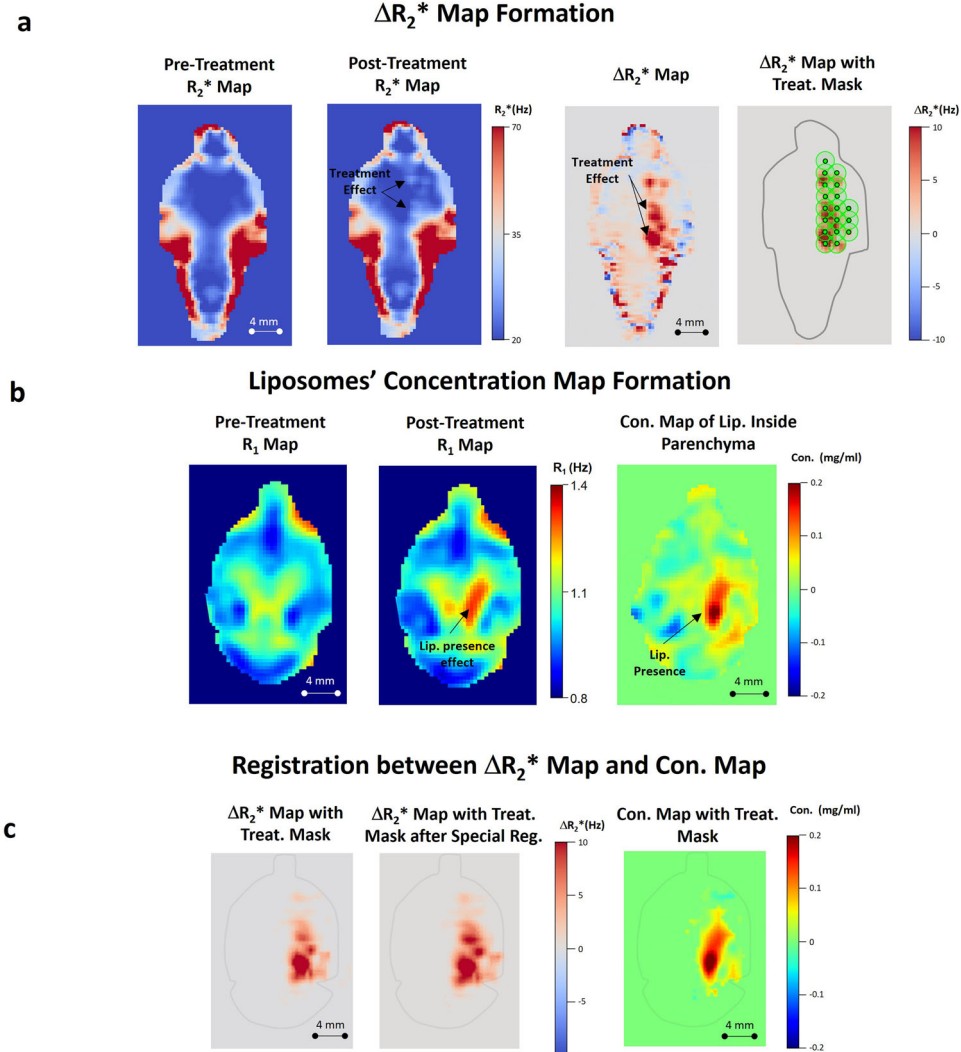

**Fig. 3 Analysis fundamentals. a** Pre- and post-treatment $R_2^*$ maps (left) and $\Delta R_2^*$ map, with and without a treatment mask (right). The white marks in the post-treatment $R_2^*$ map (indicated by arrows) and the red marks in $\Delta R_2^*$ maps indicate the treatment-related changes. **b** Pre- and post-treatment $R_1$ maps (left) and Gd-liposomal concentration map (right); arrows indicate liposome presence. **c** $\Delta R_2^*$ map with treatment area mask, after registration to the Gd-liposomal concentration map orientation (left). $\Delta R_2^*$ map after diffusion mimicking registration to the liposome concentration map (middle); note the $\Delta R_2^*$ distribution radial expansion. Gd-liposomal concentration map with treatment area mask (right).

44-nm liposomes in the $\Delta R_2^*$ range of 0–4 Hz (highest $p$ value $= 1.38 \times 10^{-21}$) and between 44-nm liposomes and 18-nm or 24-nm liposomes in the $\Delta R_2^*$ range of 10–16 Hz (highest $p$ value $= 1.43 \times 10^{-10}$, Fig. 5b). One-way ANOVA also detected significant differences between each liposome population $\Delta R_2^*$ threshold region (highest $p$ value $= 0$).

Finally, to test the statistical significance of the dependence of the liposome parenchymal concentration and $\Delta R_2^*$, a one-way ANOVA inside each liposome population revealed statistically significant differences between the parenchymal concentrations of adjacent $\Delta R_2^*$ 2-Hz bins; for 18-nm liposomes, a significant difference was detected in all the presented ranges (i.e., $\Delta R_2^* = 0$–16 Hz, highest $p$ value $= 0.046$), for 24-nm liposomes, a significant difference was detected between $\Delta R_2^* = 0$–14 Hz (highest $p$ value $= 0.024$), and for 44-nm liposomes, a significant difference was detected between $\Delta R_2^* = 6$–12 Hz (highest $p$ value $= 0.0125$).

## Discussion

The present study aimed to improve the performance, efficacy, and interpretation of data from future clinical studies of MRgFUS-induced BBB-opening using the Exablate Neuro system. This study explored the intriguing relationship between two important physical indices: the $R_2^*$ parameter[42] and the quantity of a therapeutic agent surrogate able to reach the treated area in the brain following MRgFUS BBB-opening treatment. Utilization of a standard MR parameter such as $R_2^*$, which can be measured in practice during the treatment without the need for advanced MR acquisition techniques, to predict the effective therapeutic dose in the treated area is extremely important for the rapid adoption of a therapeutic technique with an effective outcome.

The $\Delta R_2^*$ parameter is usually used as a safety parameter during BBB-opening treatments[27] to prevent vascular damage in the treatment area. Here, we found that the $\Delta R_2^*$ parameter can also be utilized to predict the effectiveness of the BBB-opening treatment by serving as an index of the delivery of molecules with sizes similar to those of drugs used to treat various neurologic problems[6,43–45]. Changes in $R_2^*$ values are directly related to the total amount of deoxyhemoglobin in the tissue[35]. Because hemoglobin is a large protein (64 kDa tetramers)[36] that can be extravasated due to hemolysis[37] without necessarily causing damage to blood vessels during the BBB-opening treatments[34], it

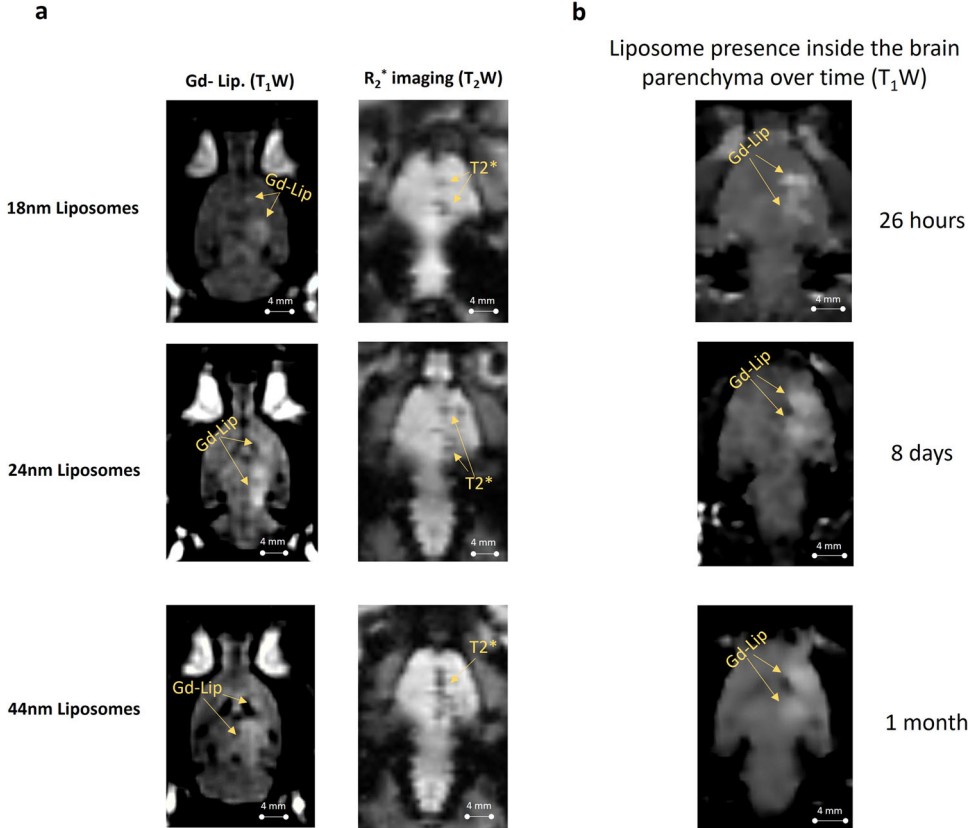

**Fig. 4 Liposome presence inside the brain parenchyma. a** Qualitative observation without registration alignment of different liposome populations' Gd signal ($T_1$-weighted imaging, left, white areas) and its treatment-related qualitative $R_2^*$ changes ($T_2$-weighted imaging, right, dark areas); although no registration was performed, it noticeable that the liposome location correlated with the local $R_2^*$ changes. **b** 44-nm liposomes presence inside the brain parenchyma over a period of 1 month. $T_1$-weighted imaging parameters: FSE, TR: 500 ms, TE: 22 ms, ET: 4. $T_2$-weighted imaging parameters: GRE, TR: 620 ms, TE: 38 ms; slice thickness: 2 mm, FOV: 18 cm, grid resolution: 224 × 224 were same for both the $T_1$- and $T_2$-weighted images.

is logical that changes in $R_2^*$ values correlate with the concentration of large molecules in the brain parenchyma.

In this study, we used various sized liposomes to model the delivery of a variety of therapeutic molecules. The liposomes incorporated Gd ions[46] to make them traceable and measurable by MRI and were produced in three different clinically relevant sizes[6,43–45]. The choice of PEG-based liposome nanoparticles was related to their clinical relevance as nano-carriers as well as to the slow metabolic nature of these molecules[41], which can stabilize their parenchymal concentration following BBB transfer. Indeed, the parenchymal presence of Gd-liposomes was detected several hours after injection and remained stable for at least 1 month following BBB closure (Fig. 4b), when rats were killed. These molecules may linger for even longer periods of time, constituting a potential platform for sustained drug delivery in the brain.

Our analyses included calculations of $R_2^*$ and $R_1$ maps from pre- and post-treatment MR data with which we calculated treatment-related $\Delta R_2^*$ and Gd-liposomal concentration maps. We then attempted to correlate $\Delta R_2^*$ values with Gd-liposomal concentrations, considering the liposome diffusion process[47] following their entrance into the brain parenchyma (Fig. 3). The results revealed statistically different BBB transfer dynamics for the different sized liposomes as a function of $\Delta R_2^*$ (Fig. 5); the smaller the nanoparticle, the greater its parenchymal concentration for the same $\Delta R_2^*$ value. In addition, the analysis demonstrated liposome size-dependent concentration plateau regions, where the parenchymal concentrations were $\Delta R_2^*$ independent; the larger the nanoparticle, the higher its $\Delta R_2^*$ threshold dependence. These dynamics raise questions regarding the use of small MRI contrast agents that can

pass through the BBB without evidence of $\Delta R_2^*$[27] to quantify general BBB-opening[48–50] when the ability and flexibility to control the drug concentration in the treatment area is critical for treatment success. Slow wash-in and washout dynamics of small MRI contrast agents (tens of minutes[51–53]) is another concern when multiple administrations of these molecules are required to achieve the final therapeutic goal in each treated area. Taking into account our study results, the safety concerns, and the total treatment length when administering multiple small MRI contrast agents, it is clear that using such small molecules (~1 nm diameter[39,40]) may be problematic for quantifying the BBB-opening level required for large molecular drugs like monoclonal antibodies and liposomal-based drugs[6,43–45].

In the current study, $R_2^*$ imaging was performed before the liposomes were administered (see Fig. 2a for detailed treatment flow), but in a clinical setting the drug can also be administered before or during treatment due to its fast pharmacokinetics and \or treatment duration that can last tens of minutes due to the much larger treatment volumes, during which the BBB-opening of earlier treated areas can already be in its closing dynamic state; the window during which the BBB is open, therefore, informs on when drug administration must be completed. When the drug is administered before or during the treatment, it may affect the $R_2^*$ values, depending on its paramagnetic properties and its BBB transition dynamics; the effects of the administration of such drugs before or during treatment on $\Delta R_2^*$ values, per used drug, requires further investigation.

Other rodent studies revealed correlations between nano-particle concentrations and harmonic and ultra-harmonic-based

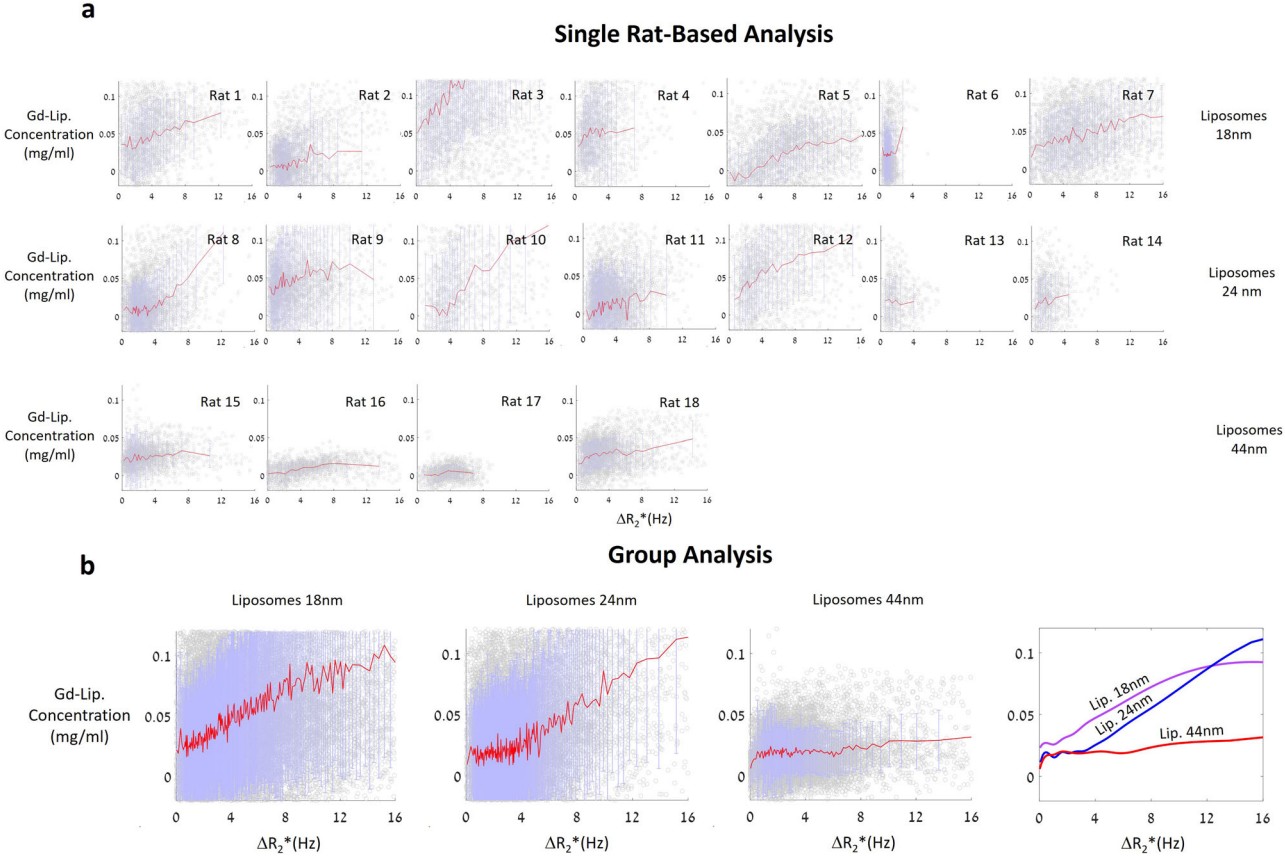

**Fig. 5 Relation between $\Delta R_2^*$ and Gd-liposome concentrations. a** Voxel-wise single rat-based analysis for different Gd-liposome populations. Gray circles are the voxel-wise data points, red curves are the means of every 100 adjacent points in the figures, and purple bars are the standard deviations. Inter-rat differences are noticeable. **b** Same analysis as in **a**, but with pooled data from all rats in the same Gd-liposome population. The right-most figure contains interpolation curves of the curves in the left panels. Each population has other characteristics: the smaller the liposome, the higher its parenchymal concentration for the same $\Delta R_2^*$ values. In addition, the larger the liposome, the higher its $\Delta R_2^*$ threshold dependence. One-way ANOVA revealed a statistically significant difference between the liposome population parenchymal concentrations within 2-Hz $\Delta R_2^*$ bins. In addition, the analysis revealed a significant difference between the liposome threshold levels, and the significant dependence of the parenchymal concentration on $\Delta R_2^*$. Error bars are ±STD.

cavitation doses[31]. In a clinical scenario, however, $\Delta R_2^*$ is the only candidate for therapeutic outcome prediction when it comes to BBB-opening. Comparison between acoustic emissions from microbubble activities in rat (Fig. 6a) and pig (Fig. 6b) reveals difficulties in capturing the changes in harmonic emissions in pigs during the BBB treatment, as (1) the second harmonic has poor transmission through the skull, and (2) the second harmonic is apparently less prominent when treating larger species, potentially owing to differences in effective blood concentrations of the ultrasound contrast agent (see "Methods" for details). The translation from acoustic feedback in small animal studies[27,31,54–56] and the BBB-opening levels to clinical BBB treatments is problematic.

The $\Delta R_2^*$ measurement is not the only MR-based technique that can be applied to evaluate BBB-opening and potentially predict BBB-permeable drug delivery efficacy. Other techniques, such as MR dynamic contrast enhancement[51–53], MR spectroscopy[57], and positron emission tomography[58,59], can contribute to BBB-opening evaluation. There are, however, limitations to these techniques. Dynamic contrast enhancement requires long measurement times (>30 min), MR spectroscopy requires high magnetic fields (>3 T) to have a sufficient signal to noise ratio, and positron emission tomography requires the use of radioactive tracers, while $\Delta R_2^*$ can be measured intraoperatively (Insightec already performs intraoperative $R_2^*$ imaging) using a

standard clinical 1.5 T MRI, which allows for real-time treatment management based on confirmed results.

The findings of the current study can be leveraged to improve the efficiency and clinical outcomes in future clinical studies of drug delivery for glioblastoma multiforme (GBM).

GBM is a highly common and lethal central nervous system cancer; it is one of the most infiltrating, aggressive, and poorly treated brain tumors with a progression-free survival of 7 months and median overall survival time from the initial diagnosis of 12–18 months[60–62].

Current therapeutic strategies, including surgery, chemotherapy, and radiotherapy, have very limited effects toward extending the life expectancy of GBM patients[60,63]. Unfortunately, the results of clinical trials[64,65] conducted in the last two decades have failed to demonstrate the efficacy of local delivery of new innovative therapeutic agents and have therefore significantly limited the advancement of new therapies for high-grade gliomas past Phase I clinical trials.

Decades of advances in MRgFUS technology now allow for performing FUS treatments through a closed cranium. These advances have had the most clinical impact in the field of movement disorders. The effectiveness of the ExAblate Neuro MRgFUS system was first demonstrated in the treatment of patients with essential tremor, and this treatment was approved by the FDA in 2016 with full Medicare coverage in all 50 US states as of July 2020. Since then, MRgFUS ablation with the

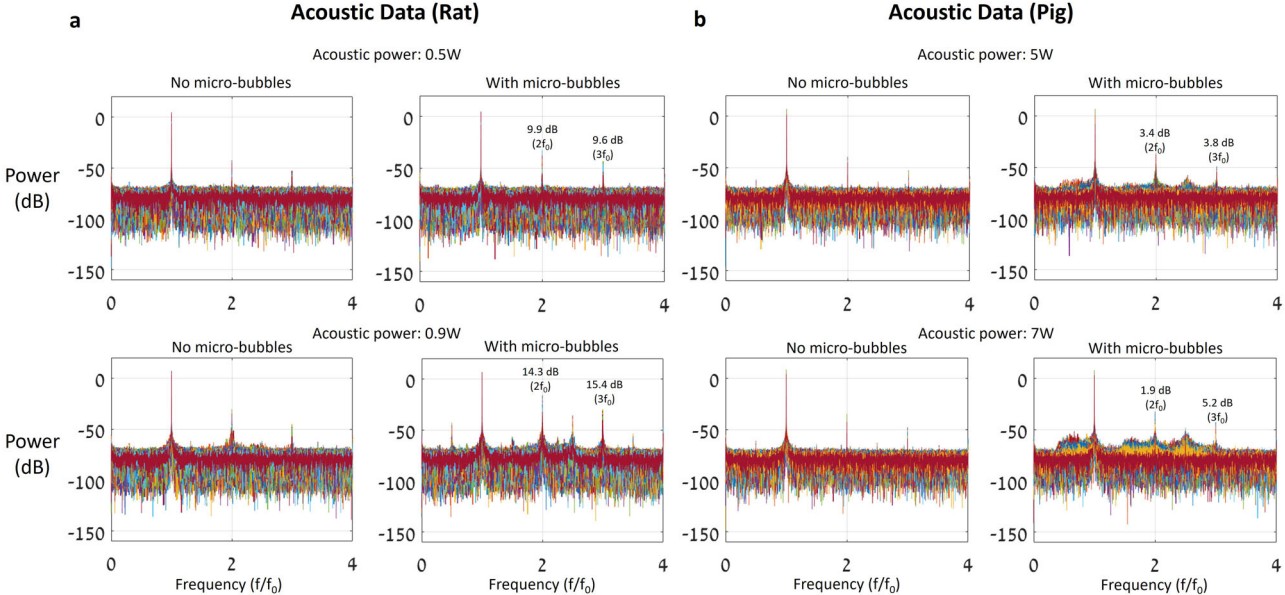

**Fig. 6 Pig and rat power spectrum data comparison.** Power spectrums of a rat and a pig, in two acoustic powers: 0.5 and 0.9 W for rat (**a**) and 5 and 7 W for pig (**b**). Each figure in **a** contains power spectrums of 490 pulses and each figure in **b** contains power spectrum of 144 pulses. The numbers in the "With microbubbles" figures show the second and third harmonic power changes compared with the baseline ("No microbubbles); the harmonic power changes in the pig were much lower despite the higher acoustic powers.

Exablate Neuro has also been approved by the FDA for Parkinson's disease (November 2021).

Due to the success of trans-skull MRgFUS treatments for movement disorders[66], oncologists have become interested in applying this technology to cancers of the central nervous system. The ability to perform noninvasive treatments with submillimeter precision is considered to be especially critical for high-grade gliomas, which are aggressive, infiltrative, and non-localized. The best strategy for treating these highly aggressive tumors is by combining localized BBB-opening with systemic administration of targeted therapy directed at infiltrative tumor cells.

McDannold et al.[27] used the ExAblate Neuro system to demonstrate actively-controlled BBB-opening that enhanced carboplatin delivery without neurotoxicity in an aggressive and infiltrative rat F98 glioma model. Based on this preclinical study, clinical trials evaluating the safety and feasibility of this technology were initiated. Mainprize et al.[67] demonstrated the safety and feasibility of MRgFUS BBB-opening with systemic chemotherapy of liposomal doxorubicin or temozolomide (TMZ) in five patients with previously confirmed high-grade glioma. Currently, several multi-site Phase I-II clinical trials are being conducted to evaluate progression-free survival and overall survival of patients with recurrent high-grade gliomas after ExAblate Neuro BBB-opening in targeted areas followed by systemic administration of carboplatin or TMZ (up to six cycles, NCT04440358, NCT04417088, NCT03712293).

Other groups report clinical data supporting the use of other devices to produce a low-intensity pulsed ultrasound to open the BBB. Chen et al.[68] used a device combining neuro-navigation and a manually operated frameless NaviFUS system (NaviFUS Inc.) to treat patients with recurrent GBM in a pilot trial. A reversible dose-dependent BBB-opening effect was observed, and safety and feasibility were established (NCT03626896). Additional studies with the NaviFUS are in progress (e.g., NCT04446416).

Several years ago, Carpentier et al.[69] reported preliminary findings in Phase 1/2a clinical trials that show safe and tolerated recurrent BBB-opening with the SonoCloud ultrasound device before treatment with carboplatin in patients with recurrent GBM

(NCT02253212); additional studies with the SonoCloud are in progress (e.g., NCT03744026 and NCT04528680)

Recently, an open-label, Phase 2, multicenter, double arm, randomized, interventional trial (NCT04614493) was initiated to evaluate progression-free survival and overall survival between the standard of care treatment (gross total surgical resection followed by radiotherapy and adjuvant TMZ) with concomitant ultrasound-mediated BBB-opening prior to TMZ administration versus standard of care alone.

In addition to the ongoing trials of small-molecule and monoclonal antibody therapies[6], future clinical trials will aim to deliver innovative therapeutic agents such as lipid nanoparticles carrying small interference RNA and clustered regularly-interspaced short palindromic repeats, which is feasible based on preclinical data performed by our group[70–72]. The current and future trials will benefit significantly from the ability to predict and quantify BBB transfer of large molecules using intraoperative $\Delta R_2^*$ measurements.

In summary, we demonstrated the ability of $\Delta R_2^*$ parameters to predict the BBB transfer of different sized molecules. This ability will enable MRgFUS BBB-opening treatments to achieve effective therapeutic outcomes. Future experiments in large animals such as pigs, as well as exploration of other types of nanoparticles besides liposomes, will evaluate $\Delta R_2^*$ prediction performance toward the eventual goal of implementation in humans.

## Methods

**Animal preparation.** Rats' experiments were performed on male Sprague Dawley rats ($n = 18$, 8 weeks old). The rats were anesthetized with a ketamine (100 mg/kg)-xylazine (10 mg/kg) mixture. A 24G cannula was inserted into the rats' tail vein. The rats' heads were completely shaved, followed by application of depilatory cream for 10 min. The pig experiment was performed on female Landrace pig (9 weeks old). The pig was anesthetized with 3 mg/kg/min of propofol and was continuously ventilated with a respirator (18 breaths per min), under 100% of $O_2$, throughout the procedure. A 20G cannula was inserted into the pig's right hind limb vein and its head was completely shaved.

The rats and the pig experiments were approved by the ethical committee of Sheba Hospital, Israel.

**FUS treatment flow**. For the BBB treatment, the rat was placed supine inside a specially designed stereotactic holder. The holder was placed on top of the MRgFUS transducer, and a MR loop coil (three-inch round coil) was placed on top of the rat's head area (Fig. 1a). The transducer was then inserted inside the MRI (1.5 T SIGNA EXCITE HD, GE) and $T_2$-weighted treatment planning images were acquired ($T_2$ Fast Spin Echo [FSE] sequence, TR: 5500 ms, TE:100 ms, ET: 12, slice thickness: 1.5 mm, FOV: 18 cm, grid resolution: 224 × 224).

The planning images were then uploaded to the Exablate Neuro 7.42 software (Insightec, Israel) designed for clinical BBB-opening treatments to define targets (15–18 spots) in the right hemisphere (cerebellum area not included, Fig. 1b); the distance between the target centers was 2 mm.

Before starting the treatment, additional multi-gradient echo (GRE) volumes were acquired for pre-treatment $R_2^*$ ($1/T_2^*$) value calculations (spoiled gradient recalled echo [SPGR] sequence, flip angle: 70°, TR: 1000 ms, slice thickness: 2 mm, TE: 5.4, 14.4, 23.4, 32.4, 41.4 ms, FOV: 18 cm, grid resolution: 224 × 224). Subsequently, the treatment was started by bolus injection into the tail vein of ~$2 \times 10^8$ MBs/kg with a 1:10 dilution in saline (200 μl/kg MBs mixture), followed by bolus injection of 200 μl saline to facilitate full administration; the microbubbles prepared in-house (see details below and Supplementary Fig. 1a for microbubble distribution). The ultrasound treatment started 20 s after the injection using a controller over sub-harmonic emissions (see ultrasound treatment control parameters). At the end of the treatment, we repeated the acquisition of multi-GRE volumes for post-treatment $R_2^*$, and treatment-related difference $\Delta R_2^*$ calculations.

Gd-liposomes at a dosage of 0.49 ± 0.04 mmol total lipid quantity/kg body weight were injected into the rat ~15 min after the treatment ended. For Gd-liposome concentration measurements (see Supplementary Fig. 1b for Gd-liposomes populations distribution), we used FSE sequences with different TRs (TR: 160, 200, 400, 800, 1400, 3000 ms, TE: 13 ms, slice thickness: 2 mm, FOV: 18 cm, grid resolution: 224 × 224). This sequence was used to calculate $R_1$s ($1/T_1$) pre- and post-treatment after Gd-liposomes injection, followed by $\Delta R_1$ calculations that were translated to Gd-liposomes concentrations using the calibration curves shown in Supplementary Fig. 2; these curves were established using rat blood with various Gd-liposome concentrations and the same FSE acquisition parameters with a single 10-mm thickness slice. The experimental flow is summarized in Fig. 2a.

**MRI analysis**. Details of the MRI analysis flow are presented in Fig. 2b. First, $R_1$ and $R_2^*$ were calculated per-voxel for pre- and post-treatment volumes. Then, linear interpolation was used to reconstruct: (1) $R_1$ and $R_2^*$ volumes, (2) the pre- and post-treatment SPGR volumes with TE of 5.4 ms and (3) FSE volumes with TR of 3000 ms, to a voxel size of 0.5 mm³, when (2) and (3) were used to find the registration transformation (rigid and then affine, SimpelElastix module[73]) between the pre- and post-SPGR and FSE volumes, respectively.

After the transformations were found, they were applied to pre-treatment $R_2^*$ and $R_1$ volumes, respectively, to calculate $\Delta R_2^*$ and $\Delta R_1$ (Fig. 3a, b, respectively); the $\Delta R_1$ volumes were converted to Gd-liposomal parenchymal concentrations using the slopes of the calibration curves (Supplementary Fig. 2) as shown in Eq. (1):

$$C_{\text{Lip}} = \frac{\Delta R_1}{\text{Clib.Slope}} \tag{1}$$

where $C_{\text{Lip}}$ is the Gd-liposomal concentration and Clib.Slope is the $R_1$ to Gd-liposomal concentration calibration slope of the injected Gd-liposomal population (i.e., 18, 24, or 44 nm, Supplementary Fig. 1b).

To focus on the region of interest in the brain, we applied brain masks on both $\Delta R_2^*$ and $\Delta R_1$ volumes to filter out voxels outside the targeted brain region (Fig. 3a, b); four rats were chosen as "model rats", for which we created SPGR (TE of 5.4 ms) and FSE (TR of 3000 s) brain masks manually and used non-rigid transformation (rigid, affine, and then B-spline transform, SimpelElastix module[73]) to apply the manually-created brain masks to each analyzed rat.

The treatment-related $\Delta R_2^*$ spatial changes were then isolated according to the sub-spot locations (Fig. 3a and Supplementary Fig. 3) and subsequently used as a mask for Gd-liposomal concentrations after registration between $\Delta R_2^*$ and Gd-liposomal concentration volumes (Fig. 3c), which was established utilizing SPGR (TE of 5.4 ms), and FSE (TR of 3000 ms) volumes after application of appropriate brain masks that facilitated the registration.

Under the assumption that liposomes undergo diffusion once they enter the brain parenchyma[47], we performed another affine registration between $\Delta R_2^*$ and Gd-liposomal concentration maps to mimic the diffusion phenomenon and determine the best correlation between the two maps in the treatment area. We then plotted the liposomal concentration as a function of the $\Delta R_2^*$ parameter voxel-wise, to evaluate the relation between the two physical parameters, at both individual and group levels.

**Liposomes and microbubble preparations**. The reagents and materials used to prepare the liposomes and microbubbles were acquired from Sigma-Aldrich Ltd, Rehovot, Israel, unless otherwise specified.

The liposomes used in our study were coupled with Gd ions to make them visible and quantifiable by the MRI scanner. Two methods were used for preparing these liposomes: (1) an ultrasound probe and extruder-based approach[46] and (2) a microfluidics-based approach[74].

The lipids used for production were: 1,2-dipalmitoyl-sn-glycero-3-phosphocholine (DPPC, C.N. 850355P), Gd-labeled lipid (DTPA-bis(stearylamide) [Gd salt], Gd-DTPA-BSA, C.N. 791268P), 1,2-distearoyl-sn-glycero-3-phosphoethanolamine-N-[methoxy(polyethylene glycol)-2000] [ammonium salt] (DSPE, C.N. 880120P), and 1,2-dipalmitoyl-sn-glycero-3-phosphoethanolamine (DPPE, C.N. P1348), with a molar ratio composition of DPPC:DTPA:DSPE:DPPE, 58.5:35.5:5.0:1.0 mol %, respectively[46].

The lipid mixture was prepared by thin-film hydration, as follows: lipids were dissolved in a 2:1 v/v ratio of chloroform: methanol at 40 °C for 2 h (chloroform ≥ 99.5%, C.N. C2432 and methanol = 99.8%, C.N. 322415), and then the solvents were evaporated in an evaporator at 50 °C under vacuum. The lipid thin-film was left in a vacuum oven overnight at 50 °C to remove residual solvent. The lipid thin-film then underwent two different production routes depending on the produced liposome size:

(1) Liposome preparation using an ultrasound probe and extruder:
The lipid thin film was rehydrated with phosphate-buffered saline (PBS, pH 7.2, C.N. 806544) for 3–5 h at 70–80 °C with stirring. The lipid solution was then sonicated at 20% power for 2 min with a sonicator probe and extruded through membrane filters with 1-μm pores (x4, C.N. 610010), 0.8-μm pores (x4, C.N. 610009), 0.4-μm pores (x4, C.N. 610007), 0.22-μm pores (x4, C.N. 610006), and 0.1-μm pores (x8, C.N. 610005) to produce type A liposomes with a peak diameter of 44-nm (diameter distribution of 49.8 ± 14.9 nm), or adding another step of 0.05-μm pores (x8, C.N. 610003) to produce type B liposomes with a peak diameter of 24 nm (diameters distribution of 29.5 ± 13.7 nm). The extruder (C.N. 610000) and membranes were manufactured by Avanti Polar Lipids, supplier: Sigma-Aldrich Ltd.

(2) Liposome preparation using microfluidics:
Alternatively, the lipid thin film was dissolved in ethanol (99.5%, C.N. 500535001, Chen Samuel Chemicals Ltd., Haifa, Israel) for 3–5 h at 60 °C while stirring (lipids solution). Another solution of PBS was stirred at 60 °C for 3–5 h (PBS solution). The type C Gd-liposomes with a peak diameter of 18 nm (diameter distribution of 20.4 ± 7.35 nm) were produced by rapid microfluidic mixing (Herringbone Mixer, Darwin Microfluidics ltd.) with a v/v ratio of 5:1 PBS solution:lipid solution. The liposome distribution was characterized by dynamic light scattering method, Zetasizer Ultra Malvern Panalytical Ltd (see Supplementary Fig. 1a).

Microbubbles were prepared as described elsewhere[75]. Microbubbles were characterized by a polydisperse size distribution. The average concentration and number-weighted mean diameter of the microbubbles were $(1.0 \pm 0.3) \times 10^{10}$ microbubbles/ml and 0.88 ± 0.283 μm, respectively; the distribution and concentration were characterized by AccuSizer® FX - Entegris, Inc. (see Supplementary Fig. 1b).

**Ultrasound treatment parameters**. All the BBB treatments were performed with the Insightec Ltd 230 kHz MRgFUS transducer (Exablate Model 4000 Type-2 system, see Fig. 1a). A pulse duration of 5 ms with a pulse repetition frequency of 1 Hz for each sub-spot was used during the treatment; the transition rate between the sub-spots was in the range of 15–18 Hz.

Acoustic emissions during sonication were recorded by two passive cavitation detectors with a resonance frequency of 110 kHz. For each detector, the recorded emissions strength of each pulse was summed over a frequency band of 115 ± 40 kHz of the power spectrum to calculate the cavitation dose (CD).The total CD for each pulse was calculated by the weighted sum of the two receivers:

$$\text{CD} = 0.5 \times \text{CD}_1 + 0.5 \times \text{CD}_2 \tag{2}$$

This CD was used to calculate the updated transducer driving power, according to the next integral controller:

$$\text{CD}_{\text{Com}} = \text{CD}_{\text{total}} \frac{n}{n_{\text{total}}} \tag{3}$$

$$e[n] = \text{CD}_{\text{Com}} - \sum_n \text{CD}[n] \tag{4}$$

$$K_I e[n] = P[n+1] \tag{5}$$

where $\text{CD}_{\text{total}}$ is the required total treatment dose per sub-spot, $n_{\text{total}}$ is the required number of pulses to be delivered for each sub-spot, $\text{CD}_{\text{Com}}$ is the treatment's desired sub-harmonic CD to be reached in a specific sub-spot location at the $n$th sonication, $\text{CD}[n]$ is the measured sub-harmonic CD during $n$th sonication in a specific sub-spot location, $e[n]$ is the error between the desired and measured total CD during $n$th sonication in a specific sub-spot location, $K_I$ [250 AU] is the controller constant, and $P[n]$ is the MRgFUS transducer power during $n$th sonication; a separate control was carried out for each sub-spot.

The total treatment mean CD per rat, per sub-spot, was in the range of 0.15–0.25 [AU], the effective acoustic power that was measured by the Exablate system was between 1–1.5 W (acoustic amplitude of 261–320 kPa, estimated from ref. [27] measurements) and the total treatment time was 150 s (single sonication treatment). The total treatment CD range and treatment time were determined in a preliminary rats experiment that explored the relation between these parameters and $\Delta R_2^*$ values, which seem to be relevant to human clinical trials[34].

**Rat and pig spectrum experiments details**. Both rat (250 g) and pig (24 kg) experiments used the same acoustic setup (Exablate Model 4000 Type-2 system transducer, $F_0 = 230$ kHz, and passive cavitation detector), when the passive cavitation detector was a homemade 500-kHz receiver (see Supplementary Fig. 4 for details). The sonication was performed to the cortex center of both animals. Nine and ten sub-sonication points were used in pig and rats, respectively. The distance between the centers of the sub-sonications was 2 mm. The duration of each pulse was 5 ms and the pulse repetition frequency for each sub-spot was 1 Hz, for both the pig and rat. Before every sonication, a bolus injection as described in the Methods section was given to the rat (see "Results" for more information); a 20-min interval separated the sonications for microbubble clearance. In the pig experiment, microbubbles ($78 \times 10^6$ MB/min) were continuously infused; the sonication started 15 min after the infusion onset time for blood concentration stabilization.

**Statistics and reproducibility**. The ANOVA statistical analysis in Fig. 5b was performed voxel-wise (hundreds to thousands of voxels for each test). The analysis was performed in Microsoft Excel 365 (Version 2111, Build 14701.20210). The error bars in Fig. 5 are ±STD.

**Analysis implementation details**. All the MR image analyses were performed in Python 3.8.5; linear interpolations: scipy.interpolate.interpn, rigid, affine and B-spline registration transformations: SimpelElastix module[73].

**Reporting summary**. Further information on research design is available in the Nature Research Reporting Summary linked to this article.

## Data availability
Source data for plots in Fig. 5 can be found in Supplementary Data 1. Other datasets generated during and/or analyzed during the current study are available from the corresponding author on reasonable request.

## Code availability
Data analysis scripts are available from the corresponding authors upon reasonable request.

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

## Acknowledgements

We thank Natalie Vaisman, Avinoam Bar-Zion, Emily Mason, and the two anonymous reviewers for their helpful comments on the manuscript.

## Author contributions

M.P., R.D.P., E.Z., Y.L., and Z.R.C. conceived and designed the study. M.P., T.B., and G.G.S.T. carried out all the experiments. M.P. and T.B. analyzed the data with R.D.P., Y.L., and Z.R.C. T.B. prepared the liposomes and microbubbles. B.S. and Y.Z. developed the SPGR multi-echo $R_2$* sequence. J.G., R.D.P., E.Z., M.S., A.W., and Z.Z. interpreted the results and edited the manuscript. Y.L. and Z.R.C. supervised the project and prepared the manuscript with M.P. and T.B.

## Competing interests

The authors M.P., T.B., G.G.S.T., J.G., B.S., Y.Z., R.D.P., E.Z. and Y.L. are employees of Insightec and declare the following competing interests: Insightec Ltd is a commercial MRgFUS company. M.S., A.W., Z.Z. and Z.R.C. declare no competing interests.
