## [Peer Review File · Communications Biology]

Reviewers' comments:

Reviewer #1 (Remarks to the Author):

Comments to author:

Results

The authors mix material that should be included in the Methods section in the Results. The first 3 pages of the Results section should all be moved to the Methods section. The Results section should clearly report what findings were made and what is shown in the Figures of the manuscript.

Indirect method of BBB opening. Why is it better than gadolinium? It would be useful to use a realistic drug (clinical drug) and show that this method performs better at predicting drug concentrations than measuring T1 signal enhancement on gadolinium.

Discussion

Was histology done to examine contribution of blood products to T2* signal, or do authors argue that there always need to be blood products that cross the BBB for large molecules to pass as well?

How long is delta R2* changed after sonications? Will injection of a drug before/during BBB opening procedures modify and confound the measurement in a clinical setting?

Can delta R2* really be used in real-time to monitor treatments (abstract text: can be measured intraoperatively for efficient real time treatment management)? Can you show thresholds for a change that correlates with small molecule passage and another one for large molecules?

Authors mention several clinical studies using ultrasound for BBB disruption, but make no mention of studies being performed with alternative devices such as neuronavigation guided or implantable ultrasound devices (NCT04528680, Navifus/Columbia).

Reviewer #2 (Remarks to the Author):

I read this manuscript with great interest as the authors lay out a novel method to analyze the parenchymal concentration of neurotherapeutics using the $\Delta R2$ parameter.

The study carries novelty and merits publication pending reviews/edits. The authors are strongly encouraged to improve the manuscript for flow and grammar.

My overarching concern is that the study, the way it is outlined in its current form, will be difficult to be reproduced by another group regardless of access to the appropriate equipment. Please describe in much greater detail the computational aspect of the study. Some detailed comments below aim at helping the authors achieve this and improve the quality of the manuscript.

Ultrasound (US) is known to disrupt the Blood Brain Barrier (BBB) through microbubbles bio-mechanical interactions  blood-brain barrier (BBB) through microbubbles' biomechanical interactions.

Please check the manuscript for minor grammar and syntax.

These technologies include direct surgical injection, intranasal delivery, active efflux transporters (AET)-targeted strategies, Tight Junction (TJ)-targeted strategies and MR-guided laser ablation (MRgLA) technique^{11,12}  please add more original citations to support the individual techniques.

Subsequently, the treatment started by injecting into the tail vein a bolus containing about 2×10^8 MBs/kg with a 1:10 dilution in saline (200 ml/kg MBs mixture), followed by 200ml saline bolus injection to facilitate full administration  Since these are in-house MBs, please mention at this place that MBs used were made in-house with details provided in M&M.

Under the assumption that liposomes undergo diffusion once they enter the brain parenchyma  please provide a reference

and the quantity of therapeutic agent that is able to reach the treated area in the brain  technically, you only used liposomes as surrogates. Please remove therapeutic agent. You could indicate that these were used in lieu of therapeutics

Micro-bubbles (MBs) were prepared as described elsewhere⁶¹  This is a reference from a book volume. There is a large volume of journal papers that could be cited here. Please replace ref 61 accordingly.

All liposomes and micro-bubbles fundamental materials acquired from Sigma-Aldrich Ltd.  All reagents for the liposomes and microbubbles were acquired by ... unless otherwise specified. Please follow a consistent scheme for suppliers where appropriate (Catalog Number/reagent, Name of Supplier, City, Country).

Pulse duration of 5msec  please use SI nomenclature. 5 ms instead of 5 msec

Please assign to each formula a separate number designation.

is the treatment's desired subharmony CD  Do you mean sub-harmonic?

Please define $\epsilon[n]$.

$P[n]$  do you mean $P(n)$?

When defining variables, please follow a sequential order as these variables appear in the text flow.

effective acoustic power was between 1-1.5W (acoustic amplitude of: 261-320 kPa)  how were the acoustic power/amplitude measured?

each sub-spot, the PID controller was worked independently of other sub-spots.  please rewrite this sentence.

US emerges to be the most promising technology  please refrain from this statement (e.g., the most). I would say something along the lines ... US emerges to be a particularly promising technology or similar ...

robust but reversible  robust and reversible are not mutually exclusive. You could say both robust/consistent/safe and reversible

BBB disruption/opening  please use either term consistently throughout the text

Finally, we believe that the suggested study creates the ideal settings for potential clinical

trials and may become a new therapeutic modality for Glioblastoma multiforme (GBM)  I would remove this section. This study is not centered on GBM. The procedure the authors describe could be used for any related BBB application (e.g., AD, GBM, etc)

The experiment flow is summarized  experimental flow

Please define PID.

The total cavitation dose for  since cavitation dose has already been defined, please replace by CD. Please do this for all other instances in the text that apply

the required total treatment dose per sub-spot, n_{total} is the required number of total pulses to be delivered for each sub-spot  how is the required total treatment dose per sub-spot estimated? How is the required number of total pulses estimated? Significantly more detail is needed.

n_{th}  please use a consistent way for n_{th} or n^{th} superscript). Either way is fine.

MR Imaging Proves to be a Predictor for Nano-particles Concentration Delivered to the Brain Parenchyma

Review summary

Dear reviewers,

The text changes inside the manuscript are marked in blue.

Reviewer #1:

Q1: The authors mix material that should be included in the Methods section in the Results. The first 3 pages of the Results section should all be moved to the Methods section. The Results section should clearly report what findings were made and what is shown in the Figures of the manuscript.

A1: We thank the reviewer for this comment. "FUS Treatment flow" and "MR Imaging Analysis" sections were moved from *Results* to *Methods* section, which led to unification of figures 2 and 3 to a single figure.

Q2: Indirect method of BBB opening. Why is it better than gadolinium? It would be useful to use a realistic drug (clinical drug) and show that this method performs better at predicting drug concentrations than measuring T1 signal enhancement on gadolinium.

A2: The reviewer indeed raised very important question in context of Gd salts (e.g. Omiscan, Gadavist and Dotarem) usage throughout the treatment.

Due to slow wash-in and wash-out dynamics of Gd salts molecules (tens of minutes, 51–53) it is very challenging to use these molecules during treatment to verify effective therapeutic outcome of human BBB treatment; inter and intra humans' brain vascular properties difference will cause different treatment areas to be sonicated several times or having different acoustic doses to obtain effective BBB treatment (i.e. sufficient drug concentration in the treatment area). This means Gd-salts will have to be injected several times during treatment to verify that the BBB treatment has reached its final therapeutic goal in all the treated areas. This approach raises safety concerns from Gd-salts administration prospective and overall treatment time duration (that in this case can last a lot of hours).

The usage of prediction tool like the ΔR_2^* can overcome this problem since R_2^* imaging can be performed Intraoperatively; In fact, Insightec already performs R_2^* imaging during its clinical trials.

Once you reached the required ΔR_2^* values in the currently treated area you can move to treat the next brain area, knowing that the therapeutic goal is reached.

Another concern regarding the usage of Gd salts is their size compared to large drug molecules. It is known that Gd salts can pass the BBB without evidence for ΔR_2^* , however we found that different size molecules have different ΔR_2^* BBB transition thresholds, which raises questions regarding the suitability of small MRI contrast agents to quantify general BBB opening.

The current study is a proof-of-concept study; we believe that similar ΔR_2^* and molecular concentration relation will also be valid for real drugs, with obvious drug dependent curves' thresholds and slopes variability.

We added references to this topic in the manuscript in the *Discussion* as follows (the added section is marked in bold):

“In addition, the analysis demonstrated liposomes size dependent concentration plateau regions, where the parenchymal concentrations were ΔR_2^* independent; the larger the nanoparticle, the higher its ΔR_2^* threshold dependence. These dynamics raise questions regarding the use of small MRI contrast agents that can pass the BBB without evidence for ΔR_2^* ²⁷, to quantify general BBB opening^{48–50} when the ability and the flexibility to control the drug concentration in the treatment area is a critical for treatment success. **Slow wash-in and wash-out dynamics of small MRI contrast agents (of tens of minutes, (of tens of minutes^{51–53}) is another concern when multiple administrations of these molecules are required to verify the final therapeutic goal of each treated area. Taking into account our study results, the safety concerns and the total treatment length of using multiple small MRI contrast agents administrations, one can understand the problem of using such small molecules (1 nm diameter^{39,40}) to quantify BBB opening level for large molecular drugs like monoclonal antibodies and liposomal-based drugs^{6,43–45}.”**

Q3: Was histology done to examine contribution of blood products to T2* signal, or do authors argue that there always need to be blood products that cross the BBB for large molecules to pass as well?

A3: We thank the reviewer for keen examination of our results. We have not done histology to examine the contribution of blood products to R_2^* . Yet, while we know that when R_2^* appears there is a chance to have blood products in the tissue, there is evidence that this is not necessarily the case (see ref 34). Anyway, no visible impairment was observed in the behavior of the rats after treatment in our study.

We do not argue that large molecules will not cross the BBB without R_2^* , we do argue that for the MRI-based treatment strategy which is used today in patients (more than 400 treatments so far), treatment effectiveness can be controlled according to R_2^* values, due to correlation between R_2^* values and large molecules concentrations.

This subject is already addressed in the *Introduction*:

Although ΔR_2^* is typically used as a safety measure to detect treatment related petechiae during BBB opening procedures^{27,34}, the utility of ΔR_2^* is not limited to measurements of vascular damage. The results of this study indicate that ΔR_2^* values have a clear dependence on parenchymal concentrations of three clinically relevant sizes of liposomes that have been

conjugated to Gd ions to make them visible and quantifiable in the MRI environment; working in ΔR_2^* mode does not have to be accompanied by vascular damage³⁴.

and in the *Discussion* as follows (additions are marked in bold):

“The ΔR_2^* parameter is usually used as a safety parameter during BBB opening treatments²⁷ to **prevent vascular damage in the treatment area**. Here, we found it can also **control the BBB opening treatment effectiveness by serving** as a BBB opening index to predict the delivery of molecules with sizes similar to those of drugs used in the treatments of various neurological problems^{6,43–45}. Changes in R_2^* values are directly related to the total amount of deoxyhemoglobin in the tissue³⁵. Since hemoglobin is a large protein (64kDa tetramers)³⁶ that can be extravasated due to hemolysis³⁷ without **necessarily** causing damage to blood vessels during the BBB treatments³⁴, it is logical that changes in R_2^* values will be correlated to concentration of large molecules. “

Q4: How long is delta R2* changed after sonications? Will injection of a drug before/during BBB opening procedures modify and confound the measurement in a clinical setting?

A4: We acknowledge the reviewer for paying attention to our study details. We performed the R_2^* imaging 5 minutes after the treatment and explored its dynamics in some rats during the first hour after the treatment, without notice any changes. The liposomes had no effect on R_2^* values, as it was administered after the R_2^* imaging.

In clinical setting as noted by the reviewer the drug can also be administered before\during the treatment, due to fast pharmacokinetics of the administered drugs and\or treatments duration that can last tens of minutes because of much larger treatment volumes, during which the BBB opening of early treated areas can already be in closing process; the last does not allow to wait for the treatment end for drug administration. When the drug is administered before\during the treatment, its effect on R_2^* values depend on its paramagnetic properties. Future research will be required to examine the effect of paramagnetic drugs with pre or during treatment administration on ΔR_2^* parameter, per used drug.

Nevertheless, it should be noted that large drug molecules can have much slower BBB transition dynamics than that of small molecules like Gd-salts molecules. For example, the Gd-liposomes in our experiment were detected several hours after their injection (it was injected 15 minutes after the BBB treatment) compared to Gd-salts molecules that are detectable after minutes (ref 38); drug molecules with slow BBB transition dynamics can have less effect on ΔR_2^* parameter.

This question is addressed in the *Discussion* as follows:

The current study performed R_2^* imaging before the liposome’s administration (see Fig. 2a for detailed treatment flow), however in clinical setting the drug can also be administered before\during the treatment, due to its fast pharmacokinetics and\or treatments duration that can last tens of minutes because of much larger treatment volumes, during which the BBB opening of early treated areas can already be in its closing dynamic state; the last does not allow to wait for the treatment end for drug administration. When the drug is administered before\during the treatment, it may have effect on R_2^* values, depends on its paramagnetic properties and its BBB

transition dynamics; using such drugs with pre or during treatment administration will require additional exploration about its effect on ΔR_2^* parameter, per used drug.

Q5: Can delta R2* really be used in real-time to monitor treatments (abstract text: can be measured intraoperatively for efficient real time treatment management)?

A5: We thank the reviewer for rising this question. In fact, Insightec already is doing real-time qualitative R_2^* imaging during its current clinical trials and developed a capability to perform quantitative real-time ΔR_2^* imaging. All this is possible due to MRI compatibility of Insightec brain systems, including the brain system that is used for clinical BBB opening procedures (Exablate Model 4000 Type-2 system), which allows to perform the BBB treatment inside the MRI, while performing MR scans of the treated area.

In the *Discussion* we added a small reference to Insightec's capability to perform intraoperative R_2^* imaging (additions are marked in bold):

However, DCE requires long measurement times (above 30 minutes), MRS requires high magnetic fields ($> 3T$) to have sufficient signal to noise ratio, and PET requires the usage of radioactive tracers, while ΔR_2^* can be measured intraoperatively (**Insightec already perform intraoperative R_2^* imaging**) using a standard clinical 1.5T MRI, allowing for real time treatment management based on confirmed results.

Q6: Can you show thresholds for a change that correlates with small molecule passage and another one for large molecules?

A6: The answer to the reviewer's question can be found in figure 5b (Figure 6b in the original version). In this figure, size dependent ΔR_2^* thresholds are seen between the different liposomal populations ($\Delta R_2^* = 1.16, 3.6$ and 6.22 Hz, for 18, 24 and 44nm liposomes, respectively).

Q7: Authors mention several clinical studies using ultrasound for BBB disruption, but make no mention of studies being performed with alternative devices such as neuronavigation guided or implantable ultrasound devices (NCT04528680, Navifus/Columbia).

A7: We thank the reviewer for paying our attention to this matter and hence we expanded the clinical studies section to also include devices of other companies in the field.

This subject is addressed in the *Discussion* as follows (the added section is marked in bold):

“Other groups showed clinical data supporting the use of other devices of a low-intensity pulsed ultrasound to open the BBB. Chen et al⁶⁸ used a device combining neuronavigation and a manually operated frameless (NaviFUS system, NaviFUS Inc.) to treat rGBM patients in a pilot trial. Reversible dose-dependent BBB-opening effect was observed, while safety and feasibility were established (NCT03626896). Additional studies with NaviFUS are in progress (e.g. NCT04446416).

Carpentier et al.⁶⁹ reported several years ago preliminary findings in phase 1/2a clinical trials that show safe and tolerated recurrent BBB opening with SonoCloud ultrasonic device

before treatment with carboplatin in patients with recurrent GBM (NCT02253212); additional studies with SonoCloud are in progress (e.g. NCT03744026 and NCT04528680)

Recently, an open-label, Phase 2, multicenter, double arm, randomized, interventional trial (NCT04614493) is being conducted evaluating PFS and OS between the standard of care treatment (gross total surgical resection followed by radiotherapy and adjuvant TMZ) with concomitant ultrasound BBB opening prior to TMZ administration, versus standard of care alone.”

Reviewer #2:

Q1: I read this manuscript with great interest as the authors lay out a novel method to analyze the parenchymal concentration of neurotherapeutics using the ΔR_2 parameter. The study carries novelty and merits publication pending reviews/edits.

A1: We thank the reviewer for acknowledging the potential significance of this work.

Q2: The authors are strongly encouraged to improve the manuscript for flow and grammar. Please check the manuscript for minor grammar and syntax.

Particular grammar and style comments:

- **Each sub-spot, the PID controller was worked independently of other sub-spots.  please rewrite this sentence.**
- **robust but reversible  robust and reversible are not mutually exclusive. You could say both robust/consistent/safe and reversible**
- **BBB disruption/opening  please use either term consistently throughout the text**
- **The experiment flow is summarized  experimental flow**
- **The total cavitation dose for  since cavitation dose has already been defined, please replace by CD. Please do this for all other instances in the text that apply**
- **nth  please use a consistent way for nth or n(th superscript). Either way is fine**
- **When defining variables, please follow a sequential order as these variables appear in the text flow.**
- **Please assign to each formula a separate number designation.**
- **Pulse duration of 5msec  please use SI nomenclature. 5 ms instead of 5 msec**
- **is the treatment's desired subharmony CD  Do you mean sub-harmonic?**
- **P[n]  do you mean P(n)?**

A2: We thank the reviewer for bringing this matter to our attention. Hence we corrected manuscript's grammar inaccuracies, writing style problems and also corrected the manuscript flow; "FUS Treatment flow" and "MR Imaging Analysis" sections were moved from results to methods section, which led to unification of figures 2 and 3 to a single figure.

Q3: My overarching concern is that the study, the way it is outlined in its current form, will be difficult to be reproduced by another group regardless of access to the appropriate equipment. Please describe in much greater detail the computational aspect of the study. Some detailed comments below aim at helping the authors achieve this and improve the quality of the manuscript.

A3: We appreciate the reviewer's recommendations to improve the manuscript quality. We therefore elaborated the methods and discussion sections.

The performed changes are presented below.

Q4: These technologies include direct surgical injection, intranasal delivery, active efflux transporters (AET)-targeted strategies, Tight Junction (TJ)-targeted strategies and MR-guided laser ablation (MRgLA) technique^{11,12}  please add more original citations to support the individual techniques.

A4: We thank the reviewer for this comment. Additional references were added to the discussed above BBB opening technologies in the introduction (the references are marked in bold):

“US is not the only player in the field and additional technologies have been effective at overcoming the BBB. These technologies include direct surgical injection¹¹⁻¹³, intranasal delivery¹⁴⁻¹⁶, active efflux transporters (AET)-targeted strategies¹⁷⁻¹⁹, Tight Junction (TJ)-targeted strategies^{20,21} and MR-guided laser ablation (MRgLA) technique^{22,23}.”

Q5: Subsequently, the treatment started by injecting into the tail vein a bolus containing about 2*10⁸ MBs/kg with a 1:10 dilution in saline (200 ml/kg MBs mixture), followed by 200ml saline bolus injection to facilitate full administration  Since these are in-house MBs, please mention at this place that MBs used were made in-house with details provided in M&M.

A5: We thank the reviewer for this comment. This was addresses in methods as follows (the changes below are marked in bold):

“Subsequently, the treatment started by injecting into the tail vein a bolus containing about 2*10⁸ MBs/kg with a 1:10 dilution in saline (200 µl/kg MBs mixture), followed by 200µl saline bolus injection to facilitate full administration; **the MBs prepared in-house (see details below and Supplementary Fig. 1a for MBs distribution).**”

Q6: Under the assumption that liposomes undergo diffusion once they enter the brain parenchyma  please provide a reference

A6: We thank the reviewer for this comment. A reference to the diffusion phenomena of liposomes (of size of 50-200 nm) inside different media was provided (the reference is marked in bold):

In discussion:

We then attempted to correlate ΔR_2^* values and Gd-liposomal concentrations, considering the diffusion process of liposomes⁴⁷ following their brain parenchyma entrance (Fig. 4).

And in methods:

Under the assumption that liposomes undergo diffusion once they enter the brain parenchyma⁴⁷, we performed another affine registration between ΔR_2^* and Gd-liposomal concentration signals to mimic the diffusion phenomenon and find the best correlation between the two signals in the treatment area.

And the reference is:

47. Rusu, L., Lumma, D. & Rädler, J. O. Charge and size dependence of liposome diffusion in semidilute biopolymer solutions. *Macromolecular bioscience* **10**, 1465–1472 (2010).

Q7: and the quantity of therapeutic agent that is able to reach the treated area in the brain  technically, you only used liposomes as surrogates. Please remove therapeutic agent. You could indicate that these were used in lieu of therapeutics

A7: We thank the reviewer for this comment. This sentence in the discussion was slightly changed as follows (the change is marked in bold):

This study was the first to explore the intriguing relation between two important physical indices: the R_2^* parameter⁴² and the quantity of therapeutic agent **surrogate** that is able to reach the treated area in the brain following MRgFUS BBB opening treatment.

Q8: Micro-bubbles (MBs) were prepared as described elsewhere⁶¹  This is a reference from a book volume. There is a large volume of journal papers that could be cited here. Please replace ref 61 accordingly.

A8: We thank the reviewer for this comment. We changed the book reference in methods as follows (the change is marked in bold):

Micro-bubbles (MBs) were prepared as described elsewhere⁷⁵.

And the reference is:

75. Langeveld, S. A. *et al.* Ligand Distribution and Lipid Phase Behavior in Phospholipid-Coated Microbubbles and Monolayers. *Langmuir* **36**, 3221–3233 (2020).

Q9: All liposomes and micro-bubbles fundamental materials acquired from Sigma-Aldrich Ltd.  All reagents for the liposomes and microbubbles were acquired by ... unless otherwise specified. Please follow a consistent scheme for suppliers where appropriate (Catalog Number/reagent, Name of Supplier, City, Country).

A9: We thank the reviewer for this comment. We addressed the study reagents details in methods as follows (changes are marked in bold):

The reagents and materials for the liposomes and micro-bubbles preparation were acquired from Sigma-Aldrich Ltd, Rehovot, Israel, unless otherwise specified.

The liposomes used in our study were coupled to Gd-ions to make them visible and quantifiable by the MRI scanner. Two methods were used for the preparation of these liposomes: 1) An ultrasonic probe and extruder-based approach⁴⁶ and 2) A microfluidics-based approach⁷⁴.

The lipids used for production were: 1,2-dipalmitoyl-sn-glycero-3- phosphocholine (DPPC, C.N. **850355P**), gadolinium-labeled lipid (DTPA-bis(stearylamide) [gadolinium salt], Gd-DTPA-BSA, C.N. **791268P**), 1,2-distearoyl-sn-glycero-3-phosphoethanolamine-N-[methoxy(polyethylene glycol)- 2000] [ammonium salt] (DSPE, C.N. **880120P**), and 1,2-dipalmitoyl-sn-glycero-3-phosphoethanolamine (DPPE, C.N. **P1348**), with a molar ratio composition of DPPC:DTPA:DSPE:DPPE, 58.5:35.5:5.0:1.0 mol %, respectively⁴⁶.

The lipid mixture was prepared by thin-film hydration, as follows: Lipids were dissolved in a 2:1 v/v ratio of chloroform: methanol at 40°C for 2h (**chloroform ≥99.5%, C.N. C2432 and methanol=99.8%, C.N. 322415**), then the solvents were evaporated in an evaporator at 50°C under vacuum. The lipid thin film was left in a vacuum oven overnight at 50°C to remove residual solvent.

The lipid thin film then underwent two different production routes, depending on the produced liposome size:

1) Liposome preparation using an ultrasonic probe and extruder:

The lipid thin film was rehydrated with phosphate-buffered saline (PBS, **pH 7.2, C.N. 806544**) for 3-5h at 70-80°C under stirring. The lipid solution was then sonicated at 20% power for 2min with a sonicator probe and extruded through membrane filters with 1µm pores (x4, **C.N. 610010**), 0.8µm pores (x4, **C.N. 610009**), 0.4µm pores (x4, **C.N. 610007**), 0.22µm pores (x4, **C.N. 610006**), and 0.1µm pores (x8, **C.N. 610005**) to produce type A liposomes with a peak diameter of 44nm (diameters distribution of 49.8±14.9nm), or adding another step of 0.05µm pores (x8, **C.N. 610003**) to produce liposomes type B with a peak diameter of 24nm (diameters distribution of 29.5±13.7nm). **The extruder (C.N. 610000) and the membranes were manufactured by Avanti Polar Lipids, supplier: Sigma-Aldrich Ltd.**

2) Liposome preparation using microfluidics:

Alternatively, the lipid thin film was dissolved in ethanol (**99.5%, C.N. 500535001, Chen Samuel Chemicals Ltd., Haifa, Israel**) for 3-5h at 60°C while stirring (lipids solution). Another solution of PBS was stirred at 60°C for 3-5h (PBS solution).

Q10:

- **Please define [n].**
- **Please define PID.**

A10: We thank the reviewer for these comments.

e[n] was defined in methods as follows (the change is marked in bold):

...,CD[n] is the measured sub-harmonic CD during nth sonication in a specific sub-spot location, **e[n] is the error between the desired and measured total CD during nth sonication in a specific sub-spot location,...**

The term PID was replaced by “Integral”, as we actually using Integral controller and not full PID controller, to control the mirco-bubbles’ sub-harmonic emissions.

This was addressed in methods as follows (the change is marked in bold):

This CD was used to calculate the updated transducer driving power, according to the next **Integral** controller:

Q11: effective acoustic power was between 1-1.5W (acoustic amplitude of: 261-320 kPa)  how were the acoustic power/amplitude measured?

A11: We thank the reviewer for this comment, as some details regarding the applied pressure and power were missing.

The applied acoustic power is actually reported throughout the sonication by Insightec exblate system. The system electrical power is calibrated once to the acoustic power using radiation force balance (see *IEC 62555:2013 Ultrasonics - Power measurement - High intensity therapeutic ultrasound (HITU) transducers and systems*) and therefore conversion from electrical power to acoustic power can be performed at any given moment within the sonication.

On the other hand, the pressure amplitudes were estimated from measurement of ref. 27, which also used the Exablate Model 4000 Type-2 system in their experiments.

This was addressed in the manuscript as follows (changes are marked in bold):

The total treatment average CD per rat, per sub-spot, was in the range of 0.15-0.25 [AU], **the effective acoustic power that was measured by the Exablate system was between 1-1.5W (acoustic amplitude of: 261-320 kPa, estimated from Ref. 27 measurements)** and the total treatment time was 150 sec (single sonication treatment).

Q12: US emerges to be the most promising technology  please refrain from this statement (e.g., the most). I would say something along the lines ... US emerges to be a particularly promising technology or similar...

A12: We thank the reviewer for this comment. The change in the introduction section was addressed as follows (the change is marked in bold):

However, US emerges to be **particularly** promising technology as it is non-invasive, has a millimetric scale level of precision and causes robust/consistent/safe and reversible BBB opening^{24,25}.

Q13: Finally, we believe that the suggested study creates the ideal settings for potential clinical trials and may become a new therapeutic modality for Glioblastoma multiforme (GBM)  I would remove this section. This study is not centered on GBM. The procedure the authors describe could be used for any related BBB application (e.g., AD, GBM, etc)

A13: We totally agree with the reviewer's view point that this study is not centered on GBM and hence changed the sentence in the introduction section as follows (changes are marked in bold):

Finally, we believe that the suggested study creates the ideal settings for potential clinical trials and may become a new therapeutic modality for **a variety of neurological disorders such as brain tumors, neurodegenerative diseases, and cognitive disorders.**

Q14: the required total treatment dose per sub-spot, ntotal is the required number of total

pulses to be delivered for each sub-spot  how is the required total treatment dose per sub-spot estimated? How is the required number of total pulses estimated? Significantly more detail is needed.

A15: The reviewer raised an important question. The total treatment CD range and the treatment time were determined based on a preliminary rats' experiments that explored the relation between these parameters (CDs and total treatment time) and ΔR_2^* values. The results of these preliminary experiments guided us which sonication parameters to use during the current study to obtain ΔR_2^* values that are relevant to what can be seen in human clinical trials (see Ref 34).

This subject was addressed in *Method* section as follows (the addition is marked in bold):

The total treatment average CD per rat, per sub-spot, was in the range of 0.15-0.25 [AU], the effective acoustic power that was measured by the Exablate system was between 1-1.5W (acoustic amplitude of: 261-320 kPa, estimated from Ref. 27 measurements) and the total treatment time was 150 sec (single sonication treatment). **The total treatment CD range and the treatment time were determined in a preliminary rats' experiments that explored the relation between these parameters and ΔR_2^* values, which seem to be relevant to human clinical trials³⁴.**

REVIEWERS' COMMENTS:

Reviewer #2 (Remarks to the Author):

The authors appear to have address the comments of the reviewers, but the article is still not nearly ready for publication. I propose that the authors have the manuscript professionally edited and then submit to reviewers.

Reviewer #3 (Remarks to the Author):

I thank the authors for revising carefully their manuscript. This work bears merits for publication. A few minor edits have been identified that would improve the quality of the manuscript:

However, US emerges to be particularly promising  However, US emerges to be a particularly promising

and causes robust/consistent/safe and reversible BBB opening 24,25.  and causes robust, safe, and reversible BBB opening

T1W, R2* and Multi-echo R2* MR imaging were acquired before and after BBB treatment  T1w, R2*, ...

Chen et al68  et al. (please correct all occurrences in the text)

Please define rGBM

BBB-opening  please be consistent. You can use either one BBB opening or BBB-opening

Magnetic Resonance Imaging Analysis Predicts Nanoparticle Concentration Delivered to the Brain Parenchyma

Review 2 summary

Dear reviewers,

Thank you very much for all your comments; we believe that the corrections we made have significantly improved the quality of the manuscript.

Your comments were addressed in the text as follows:

The manuscript was edited, and the grammatical issues pointed out by the reviewers were addressed by changes in the text.

Specifically, regarding comments of reviewer 3:

We changed “US emerges to be...” to “US has emerged as a particularly...” (the reviewer’s suggestion was not used, but the grammar of the original sentence was corrected).

We changed "robust/consistent/safe..." to "robust, consistent, safe and reversible....".

We changed T1W to T1w and T1-weighted throughout.

The references were all formatted correctly and periods added after "et al" throughout

We spelled out “rGBM” to recurrent GBM throughout, therefore “rGBM” was not defined.

We ensured consistency of “BBB-opening” throughout (with a hyphen).

Sincerely,

The authors